# Role of PFKFB3 and PFKFB4 in Cancer: Genetic Basis, Impact on Disease Development/Progression, and Potential as Therapeutic Targets

**DOI:** 10.3390/cancers13040909

**Published:** 2021-02-22

**Authors:** Krzysztof Kotowski, Jakub Rosik, Filip Machaj, Stanisław Supplitt, Daniel Wiczew, Karolina Jabłońska, Emilia Wiechec, Saeid Ghavami, Piotr Dzięgiel

**Affiliations:** 1Department of Histology and Embryology, Wroclaw Medical University, 50-368 Wroclaw, Poland; krzysztof.kotowski@student.umed.wroc.pl (K.K.); karolina.jablonska@umed.wroc.pl (K.J.); 2Department of Pathology, Pomeranian Medical University, 71-252 Szczecin, Poland; jakubrosikjr@gmail.com (J.R.); machajf@gmail.com (F.M.); 3Department of Genetics, Wroclaw Medical University, 50-368 Wroclaw, Poland; st.supplitt@gmail.com; 4Department of Biochemical Engineering, Wroclaw University of Science and Technology, 50-370 Wroclaw, Poland; daniel.wiczew@pwr.edu.pl; 5Laboratoire de physique et chimie théoriques, Université de Lorraine, F-54000 Nancy, France; 6Department of Biomedical and Clinical Sciences (BKV), Division of Cell Biology, Linköping University, Region Östergötland, 581 85 Linköping, Sweden; ewiechec@gmail.com; 7Department of Otorhinolaryngology in Linköping, Anesthetics, Operations and Specialty Surgery Center, Region Östergötland, 581 85 Linköping, Sweden; 8Department of Human Anatomy and Cell Science, Rady Faculty of Health Sciences, Max Rady College of Medicine, University of Manitoba, Winnipeg, MB R3E 0J9, Canada; 9Research Institute in Oncology and Hematology, Cancer Care Manitoba, University of Manitoba, Winnipeg, MB R3E 0V9, Canada; 10Department of Physiotherapy, Wroclaw University School of Physical Education, 51-612 Wroclaw, Poland

**Keywords:** PFKFB3, PFKFB4, PFK-2, 6-phosphofructo-2-kinase/fructose-2,6-bisphosphatase, 3PO, PFK-158, PFK-15, autophagy, angiogenesis, cancer

## Abstract

**Simple Summary:**

Recently, our understanding of PFK-2 isozymes, particularly with regards to their roles in cancer, has developed significantly. This review aims to compile the most crucial achievements in this field. Due to the prevailing number of recent studies on PFKFB3 and PFKFB4, we mainly focused on these two isozymes. Here, we comprehensively describe the discoveries and observations to date related to the genetic basis, regulation of expression, and protein structure of PFKFB3/4 and discuss the functional involvement in tumor progression, metastasis, angiogenesis, and autophagy. Furthermore, we highlight crucial studies on targeting PFKFB3 and PFKFB4 for future cancer therapy. This review offers a cutting-edge condensed outline of the significance of specific PFK-2 isozymes in malignancies and can be helpful in understanding past discoveries and planning novel research in this field.

**Abstract:**

Glycolysis is a crucial metabolic process in rapidly proliferating cells such as cancer cells. Phosphofructokinase-1 (PFK-1) is a key rate-limiting enzyme of glycolysis. Its efficiency is allosterically regulated by numerous substances occurring in the cytoplasm. However, the most potent regulator of PFK-1 is fructose-2,6-bisphosphate (F-2,6-BP), the level of which is strongly associated with 6-phosphofructo-2-kinase/fructose-2,6-bisphosphatase activity (PFK-2/FBPase-2, PFKFB). PFK-2/FBPase-2 is a bifunctional enzyme responsible for F-2,6-BP synthesis and degradation. Four isozymes of PFKFB (PFKFB1, PFKFB2, PFKFB3, and PFKFB4) have been identified. Alterations in the levels of all PFK-2/FBPase-2 isozymes have been reported in different diseases. However, most recent studies have focused on an increased expression of PFKFB3 and PFKFB4 in cancer tissues and their role in carcinogenesis. In this review, we summarize our current knowledge on all PFKFB genes and protein structures, and emphasize important differences between the isoenzymes, which likely affect their kinase/phosphatase activities. The main focus is on the latest reports in this field of cancer research, and in particular the impact of PFKFB3 and PFKFB4 on tumor progression, metastasis, angiogenesis, and autophagy. We also present the most recent achievements in the development of new drugs targeting these isozymes. Finally, we discuss potential combination therapies using PFKFB3 inhibitors, which may represent important future cancer treatment options.

## 1. Introduction

Glycolysis is an essential enzymatic process in human cell metabolism. It participates in the production of substrates that are required in multiple biochemical pathways, such as the tricarboxylic (TCA) acid cycle, pentose phosphate pathway (PPP), and fatty acids and cholesterol synthesis. In normal human cells (with the exception of red blood cells), anaerobic reactions predominate in the metabolism under reduced oxygen conditions. However, in 1927, Otto Warburg reported an essential role of glycolysis in cancer cells regardless of oxygen concentration in the tumor microenvironment [1,2,3]. This reprogramming of cancer cell metabolism is not only responsible for its aggressive growth but may also cause a beneficial decrease in Reactive Oxygen Species (ROS) generation and key metabolites for cell growth [4]. It is worth noticing that a similar shift in metabolism is found in proliferative normal cells such as lymphocytes and endothelial cells in angiogenesis [5]. In recent years, targeting key regulatory steps of glycolysis has increasingly become an area of interest among scientists. There are many reports on novel inhibitors affecting distinct molecular targets in this process [6]. Amino acid sequence alterations leading to changes in enzyme catalytic activity have been detected in numerous proteins involved in glycolysis in different types of cancer [7].

Glycolysis intensity is regulated by the activity of three physiologically irreversible enzymes: hexokinase, phosphofructokinase-1 (PFK-1), and pyruvate kinase. PFK-1 is the main rate-limiting enzyme of glycolysis and is responsible for the synthesis of fructose-1,6-bisphosphate from fructose-6-phosphate (F-6-P). Its activity is regulated by cytoplasmically localized metabolic products, such as adenosine triphosphate (ATP), adenosine diphosphate (ADP), F-6-P, and fructose-2,6-bisphosphate (F-2,6-BP) (Figure 1) [8]. Of these compounds, F-2,6-BP, a product of the reaction catalyzed by 6-phosphofructo-2-kinase/fructose-2,6-bisphosphatase (PFK-2/FBPase-2, PFKFB), is the most potent positive allosteric effector of PFK-1 [9]. PFK-2/FBPase-2 is a bifunctional enzyme responsible for the catalyzation of both the synthesis and degradation of F-2,6-BP mediated through its N-terminal domain (2-Kase) and C-terminal domain (2-Pase), respectively [10]. Of note, the active site of the 2-Kase domain has two distinct areas (the F-6-P binding loop and ATP-binding loop) essential for its function [4].

In humans, PFK-2/FBPase-2 is encoded by four different genes: PFKFB1, PFKFB2, PFKFB3, and PFKFB4 [13]. Thus far, four different PFK-2/FBPase-2 isozymes (PFKFB1, PFKFB2, PFKFB3, and PFKFB4) have been identified. Isozymes are characterized by tissue and functional specificity [14]. PFKFB1 can be found in the liver and skeletal muscle, PFKFB2 predominates in cardiac muscle, PFKFB3 is ubiquitously expressed, while PFKFB4 occurs mainly in testes [11]. The overexpression of two isozymes (PFKFB3 and PFKFB4) has been demonstrated in various solid tumors and hematological cancer cells [15,16,17].

Furthermore, due to slight differences in amino acid sequences at key sites for enzymatic activity, all of the isozymes have a different affinity for the synthesis or degradation of F-2,6-BP. Their activity is expressed as the kinase/phosphatase ratio (also termed the 2-Kase/2-Pase activity ratio) [11]. This ratio is about 4.6/1 for PFKFB4 and 730/1 for PFKFB3, while it does not exceed 2.5/1 for PFKFB1 and PFKFB2. Isoforms commonly expressed in tumors satisfy increased energetic requirements of neoplastic cells more efficaciously. Thus, glycolysis, the hallmark of malignancy, might be vulnerable to the therapy affecting only isoforms characterized by a high kinase/phosphatase ratio [10,18].

## 2. PFKFB Genes and Proteins

The four genes encoding the different isozymes of PFK-2/FBPase-2 are located on distinct chromosomes, i.e., PFKFB1—Xp11.21, PFKFB2—1q31, PFKFB3—10p14-p15, and PFKFB4—3p21-p22 [19]. Despite the fact that the core sequences of all four genes exhibit high homology and similarities in the genomic organization, PFK-2/FBPase-2 isozymes display diverse catalytic properties (kinase/phosphatase ratio). The level of bifunctionality is determined by the unique structure of PFK-2/FBPase-2. The molecular weight of both functional domains ranges from 55 kDa to 90 kDa [20]; one terminus contains the 2-Kase domain (closer to the N-terminal end) and the other the 2-Pase domain (closer to the C-terminal end), of which the post-translational activities vary among PFK-2/FBPase-2 isozymes [21,22]. The diversity of PFKB1-4 kinase/phosphatase activity reflects the enzymatic capability of adapting to different conditions, as well as the distinct synthesis, distribution, and function of isozymes in response to physiological or pathological stimuli [8,20]. PFKFB1 encodes the isoenzyme identified in fetal tissue and the liver; PFKFB2 encodes a protein expressed mainly in the heart and kidney; the product of PFKFB3 occurs in adipose tissue, the brain, and frequently in cancer cells; and PFKFB4 is almost exclusively significantly expressed in testes and tumor cells [11]. The expression of distinct isozymes and mRNAs by these four genes can be attributed to the presence of various promoters and 5′ non-coding exons [13]. Finally, mutations in PFK-2 isozymes have been detected in several cancer tissue samples, especially in endometrial cancer, colorectal cancer, and melanoma (our preliminary data, not published yet).

For the purpose of this review, the authors discuss the structure and function of PFKFB1-4 genes and their transcripts in the following section.

### 2.1. PFKFB1

PFKFB1 contains 17 exons controlled by different promoters. Four splicing variants of PFKFB1 are known: PFKFB1-201, PFKFB1-202, PFKFB1-203, and PFKFB1-204 [13,23,24,25]. Their protein products regulate glucose metabolism in non-malignant tissues but are overexpressed in cancer cells. Of note, liver transcripts contain an additional exon encoding the N-termini, which can be phosphorylated in response to glucagon, resulting in enhanced bisphosphatase and simultaneously reduced kinase activity. Thus, glucagon induces glucose synthesis in the liver without impact on other tissues [26].

### 2.2. PFKFB2

The human PFKFB2 gene contains 15 exons, of which 9 transcripts are expressed. Only four transcripts encode the full-length protein [8,27]. It is mainly expressed in the heart, brain, lungs, kidneys, and cancer cells [28]. Analysis of heart cDNA revealed that PFKFB2 is composed of 505 amino acids and has a molecular weight of ~ 58 kDa. PFKFB2 can be phosphorylated by several protein kinases, including 3-phosphoinositide-dependent kinase-1 (PDPK-1), AMP-activated protein kinase (AMPK), protein kinase A (PKA), protein kinase B (PKB; also known as Akt), mitogen-activated protein kinase 1 (MAPK-1), and p90 ribosomal S6 kinase (RSK). Activation of RSK can be observed in BRAF ^V600E^-mutated melanoma cells where phosphorylation of PFKFB2 promotes glycolytic flux and tumor growth [29]. Furthermore, studies have indicated that hypoxia and hypoxia-inducible factor 1-alpha (HIF-1α) can regulate PFKFB2 expression [20,28,30]. In gastric cancer, for example, enhanced expression of PFKFB2 is associated with increased expression of HIF-1α-dependent genes, such as *VEGF* and *SLC2A1* [28].

### 2.3. PFKFB3

PFKFB3 contains at least 19 exons, of which 7 form a variable region and 12 constitute the constant region of the gene (Figure 2B). Moreover, in the 3’ untranslated region (3’UTR) of PFKFB3 mRNA, multiple copies of AU-rich elements are observed, which determine its increased translational activity and instability [31]. The alterations within the exons of the variable region lead to the production of six different transcripts by alternative splicing [14].

The expression of PFKFB3 is regulated by various compounds; its promotor contains response elements for estrogens, progesterone, and hypoxia-inducible compounds (Figure 2A) [32]. The PFKFB3 protein, which is the product of the *PFKFB3* gene, consists of two subunits each encompassing two domains (i.e., 2-Kase kinase and 2-Pase phosphatase domain) with distinct functions. The isoenzyme encoded by the *PFKFB3* gene has the highest kinase/phosphatase ratio among all PFK-2/FBPase-2 family members and promotes increased cellular glycolytic flux [33].

### 2.4. PFKFB4

The PFKFB4 gene contains at least 14 exons and different splice variants of PFKFB4 mRNA have been found in various tissues (Figure 3B) [26,34,35]. However, every PFKFB4 variant has identical catalytic domains. The PFKFB4 protein is a bifunctional enzyme that increases the cellular level of F-2,6-BP (and thus glycolytic flux) or decreases F-2,6-BP concentration, which results in the redirection of glucose-6-phosphate (G-6-P) towards ribose-5-phosphate (R5P) and Nicotinamide adenine dinucleotide phosphate (NADPH) synthesis in the PPP [11].

### 2.5. Comparison of PFKFB1-4 Amino Acid Sequence

PFKFB1-4 family members are highly conserved proteins (see Figure 4) with a 67–74% similar identity. The core sequences are highly homologous, with over 85% of the amino acids being identical or belonging to the same class according to The International ImMunoGeneTics System (IMGT). The 2-Pase domains of all isozymes use histidine phosphatase to break down F-2,6-BP into F-6-P [36,37,38,39]. Although the mechanism has not been investigated for the human PFK-2/FBPase-2 isozyme 4 directly, the sequential similarity to other isozymes (Figure 4) and the mouse variant (96% shared identity) allow us to hypothesize that its mechanism is similar to other isozymes [40]. The catalytic mechanism of the 2-Kase domain is less studied as compared to the 2-Pase domain and is not well characterized. However, the recent characterizations of PFK-2/FBPase-2 isozyme 3 crystal structures has revealed that it is mostly based on the stability of ATP/ADP and F-6-P molecules with the hydrogen bond network [4].

Even though the 2-Kase/2-Pase catalytic core shares high sequence homology among the four isoenzymes, there are a few differences in the amino acid sequence that strongly influence the activity of the respective domains. PFKFB3 has serine in position 303 (or sometimes 302) instead of arginine compared to the other three isozymes ((Figure 4), the circle with the yellow glow) [33,38,39]. This alteration results in a decrease in the phosphatase activity, and thus favors the synthesis of F-2,6-P by over 700 times. Furthermore, PFKFB3 has a serine in position 461 (or sometimes 460), and its phosphorylation increases the ratio of the kinase/phosphatase activity to over 3000 and markedly attenuates the sensitivity of the enzyme to inhibitors [39,44]. Similar effects of the phosphorylation of serine 466 and serine 483 in PFKFB2 have also been reported ((Figure 4), the circles with the blue and red glow, respectively) [45].

### 2.6. Structural Characteristics of PFKFB 1-4

In addition to having similar amino acid sequences (see Section 2.5), the four isozymes are highly comparable structurally as well. In Figure 5, the structures of the 2-Kase and 2-Pase domains of PFK-2/FBPase-2 isozymes 1-4 are compared. Using PyMOL software, an average root mean square deviation (RMSD) was measured between protein backbones (1.2 Å). The highest RMSD was between pairs of PFKFB2 and other structures (about 1.7 Å on average between PFKFB2 and the other isozymes) and was significantly lower between pairs of other isozymes (about 0.7 Å on average between pairwise combinations of PFKFB1 and PFKFB3, PFKFB4) [46]. Since the RMSD is a measure for the overlap in structure at the level of X-ray resolution (except for PFKFB4, which is a homology model), we can safely assume that the four isozymes are structurally very similar.

In these proteins, three distinct pockets could be identified (see Figure 6): two in the 2-Kase domains (for ATP and F-6-P) and one in the 2-Pase domains (for F-2,6-P). Pockets of ATP are also often occupied by ADP in crystallographic structures [4]. Furthermore, PFK-2/FBPase-2 inhibitors bind to the ATP/ADP pocket or to the F-6-P pocket, decreasing the glucose flux through the inhibition of F-2,6-P formation. Therefore, the ATP pocket is considered a relevant drug design target.

In the case of PFKFB2, it was found that citrate (Figure 6B), a TCA cycle byproduct, can bind to the ATP pocket in the 2-Kase domain and inhibit kinase activity [37].

PFKFB3 is distinguished by a unique β-hairpin element formed by amino acids (4-15 residues) close to the N-terminal end. This structure exclusively occurs in this isozyme and interacts with the 2-Pase domain, leading to conformational rotation and reduced phosphatase activity. This could be another possible explanation for the relatively high 2-Kase/2-Pase activity ratio of PFKFB3 [4].

### 2.7. Regulation of PFKFB Expression

PFKFB3 and PFKFB4 expression levels (and, as a result, glucose-related intracellular processes) are regulated by several molecular pathways, including those closely linked to oncogenic signaling.

#### 2.7.1. Ras-Dependent Regulation of PFKFB Expression

PFKFB3 is involved in the Ras signaling pathway, which is considered a regulator of glucose metabolism in cancer [50]. Ras-transformed cells are characterized by an increased glycolytic flux into lactate [51]. Moreover, it has been shown that the increased levels of PFKFB isozymes is highly related to hypoxic microenvironmental conditions in a HIF-1α-dependent manner, which is responsible for their expression regulation [28]; this mechanism has been observed in various cancers [52]. Interestingly, in a study by Blum et al. (2005)*,* inhibition of Ras signaling in glioblastoma caused a reduction in HIF-1α expression and, consequently, down-regulation of PFKFB3 and glycolysis, resulting in cell death [53]. In contrast, genomic deletion and siRNA silencing of PFKFB3 suppressed the growth of Ras-activated fibroblasts in athymic mice [50].

#### 2.7.2. mTOR-Dependent Regulation of PFKFB Expression

In addition to Ras-associated signaling, activation of other oncogenic pathways contributes to the stimulation of glycolysis through PFKFB3 and PFKFB4. For example, hyperactivation of mammalian target of rapamycin (mTOR) has frequently been observed in numerous cancers [54]. Aberrant mTOR signaling promotes cell proliferation and downregulates autophagy [55]. Activation of the mTOR signaling pathway upregulates PFKFB3 expression [56], which suggests a close connection between increased glycolytic flux and cancer development.

#### 2.7.3. Steroid-Dependent Regulation of PFKFB Expression

Activation of estrogen receptor (ER) signaling [57,58], human epidermal growth factor receptor 2 (HER2) overexpression [59], and loss of p53 and PTEN [60,61] further stimulate glycolysis in a PFK-2-dependent manner. Overexpression of oncogenes, such as Myc and Src, enhances PFKFB-mediated glycolysis and purine metabolism [13,62]. In a recent study by Dasgupta et al. (2018), PFKFB4 was found to increase the activity of the oncogenic transcription factor SRC-3 (steroid receptor coactivator 3) through its phosphorylation. SRC-3 activation led to redirection of glucose metabolism to PPP and enabled purine synthesis. Blocking PFKFB4 and SRC-3 suppressed cellular growth, prevented metastasis, and reduced the concentration of nucleotides in breast cancer cells [63].

## 3. PFKFB3 and PFKFB4 in Cancer

PFK-2/FBPase-2 family members, PFKFB3 and PFKFB4 in particular, are overexpressed in numerous malignancies (Table 1). PFKFB3 is frequently found in breast cancer [35,59,64,65], colon cancer [35], nasopharyngeal carcinoma [66], pancreatic cancer [67], gastric cancer [67], and many other neoplasms. Similarly, increased transcription of PFKFB4 is observed in pancreatic cancer [67], gastric cancer [67], ovarian cancer [68], breast cancer [35,69], colon cancer [35,70] and glioblastoma [71]. The significance of PFKFB3 level has been reported in cancer cells but also in tumor-related cells such as cancer stem cells. Furthermore, lower PFKFB3 and PFK-I expression levels have been demonstrated in induced pluripotent stem (iPS) cells compared to cancer and cancer stem cells (CSCs). This distinct expression pattern of PFKFB3 may improve the timely detection of CSCs [64].

**Table 1 cancers-13-00909-t001:** Expression of PFKFB3 and PFKFB4 in various cancer types.

Isoenzyme	Cancer Type	Research Environment and the Study Material and/or Cell Line		Reference
PFKFB3	Breast cancer	HMEC, MCF-10A, SKBR3, BT-474	In vitro	O’Neal et al. [59]
		HER2+ patient samples	In vitro	Novellasdemunt et al. [72]
		MCF-7, T-47D	In vitro	Imbert-Fernandez et al. [58]
		MCF-7, T-47D, SUM159	In vitro	Ge et al. [73]
		Breast cancer patient samples, MDA-MB-231, MDA-MB-438, HUVEC	In vitro	Peng et al. [65]
	Melanoma	451LU, WM983	In vitro	Warrier et al. [74]
		A375	In vitro/in vivo	Telang et al. [75]
		DB-1, SK-MEL-5	In vitro	Mendoza et al. [76]
	Gastric cancer	MKN45, AGS, BCG823, GES-1	In vivo/in vitro	Zhu et al. [77]
		MKN45, NUGC3	In vitro	Bobarykina et al. [28]
		MKN45, NUGC3	In vitro	Minchenko et al. [67]
	Pancreatic cancer	Panc1	In vitro	Minchenko et al. [67]
		Panc1	In vitro	Bobarykina et al. [28]
		Panc1	In vitro	Yalcin et al. [78]
	Colon adenocarcinoma	Colorectal cancer patient samples, SW480, SW1116	In vivo/in vitro	Han et al. [79]
		HCT-116	In vitro	Klarer et al. [80]
		FFPE tissue samples, SW620	In vitro	Atsumi et al. [81]
	Ovarian cancer	HeyA8, HeyA8MDR, OVCAR5, OV90	In vitro	Mondal et al. [82]
	Lung cancer	LLC1, H522	In vitro	Clem et al. [83]
		H522, H1437, PC9, HCC827	In vitro	Lypova et al. [84]
	Bladder cancer	T24, HUVEC	In vitro	Hu et al. [85]
	Glioblastoma	U87	In vitro	Mendoza et al. [76]
		Glioblastoma patient samples	In vitro	Kessler et al. [86]
		Glioblastoma patient samples	In vitro	Fleischer et al. [87]
		Glioblastoma patient samples, U87	In vitro	Zscharnack et al. [88]
	Head and neck carcinoma	Cal27, FaDu, HNSCC patient samples	In vitro	Li et al. [89]
	Astrocytoma	Astrocytoma patient samples	In vitro	Kessler et al. [86]
		Astrocytoma patient samples	In vitro	Zscharnack et al. [88]
	Neuroblastoma	-	Statistical analysis	Trojan et al. [90]
	Cervical cancer	OV2008, C13	In vitro	Mondal et al. [82]
	Renal cancer	ACHN	In vitro	Lu et al. [91]
	Thyroid cancer	FFPE tissue samples	In vitro	Atsumi et al. [81]
	Osteosarcoma	U20S	In vitro	Du et al. [92]
		Osteosarcoma patient samples, Saos-2	In vitro	Zheng et al. [93]
	Acute myeloid leukemia	THP-1, OCI-AML3	In vitro	Feng et al. [56]
	Esophageal carcinoma	KYSE30, KYSE150	In vitro/statistical analysis	Liu et al. [94]
PFKFB4	Breast cancer	MDA-MB-231, T47D, breast cancer patient samples	In vitro	Gao et al. [69]
		Breast cancer patient samples	In vitro	Yao et al. [95]
		MDA-MB-231, MCF7, SUM159, MDA-MB-468, breast cancer patient samples	In vitro	Gao et al. [96]
		MDA-MB-231, MCF-7, MCF-7-ERE-MAR-Luc, MCF-10A	In vitro/in vivo	Dasgupta et al. [63]
	Ovarian cancer	SKOV3, UPN-251, OC316, OVCAR-3, A2780	In vitro	Taylor et al. [68]
	Gastric cancer	MKN45, NUGC3	In vitro	Bobarykina et al. [28]
	Pancreatic cancer	Panc1	In vitro	Bobarykina et al. [28]
	Neuroblastoma	-	Statistical analysis	Trojan et al. [90]
	Prostate cancer	PC-3, LNCaP	In vitro	Li et al. [97]
		DU145, PC-3, LNCaP	In vitro	Ros et al. [98]
	Glioblastoma	NCH421k, NCH441, NCH644	In vitro	Goidts et al. [99]
	Bladder cancer	Bladder cancer patient samples	In vitro	Yun et al. [100]
	Lung adenocarcinoma	Lung adenocarcinoma patient samples, H460	In vitro	Chesney et al. [101]

### Influence of PFKFB3 and PFKFB4 on Carcinogenesis

PFKFB3 and PFKFB4 affect carcinogenesis and cancer metabolism in a multidirectional manner. Both isozymes participate in the regulation of glucose metabolism through enhancing glycolysis and PPP. These enzymatic reactions are crucial for cancer development [11]. Increased glucose metabolism through glycolysis enables cancer cells to survive in a microenvironment with limited oxygen supply and produce lactate which acidifies the adherent tissues and thus accelerates metastatic development. On the other hand, redirection of glucose to PPP allows for the synthesis of lipids and nucleic acids essential for the growth of cancer cells. The expression of both enzymes is induced by hypoxia, thereby facilitating nonoxidative glucose-dependent energetic metabolism of the cell. PFKFB3 and PFKFB4 stimulate glucose uptake and boost glycolytic flux to cancer cells by increasing F-2,6-BP, which is a compound promoting glucose utilization by glycolysis [8]. Both proteins are directly engaged in the production of ATP and Nicotinamide adenine dinucleotide (NADH), the synthesis of nucleic acids, and thus cancer cell growth.

## 4. Proliferation, Invasiveness and Migration

In many types of cancer, higher expression of PFKFB3 or PFKFB4 correlates with shorter overall survival (OS) or a more frequent presence of metastases. As tumorigenesis depends on several alterations in the cellular metabolism which enable survival in an unfavorable environment, a high rate of glycolytic flux is observed [18]. Increase in the intracellular F-2,6-BP concentration, a marker of glycolysis [101], is detected in neoplastic cells [102].

The first indication for a role of PFKFB3 in cancer cell proliferation was reported by Atsumi et al. in 2002, who demonstrated that PFKFB3 mRNA was induced during the G1/S transition and particularly during the S cell cycle phase [81]. In line with these findings, Calvo et al. described a significant growth rate reduction after silencing PFKFB3 using siRNA in HeLa adenocarcinoma cervical cancer cells [103]. In the following years, several studies confirmed the pro-proliferative effect of PFKFB3 [14].

The majority of recent evidence points to an impact of PFKFB3 on the expression levels of cyclin-dependent kinases (Cdks) and thus cell cycle arrest (Figure 7). Yalcin et al. (2009) reported that ectopic expression of PFKFB3 led to the upregulation of some Cdks, including Cdk-1, Cdc25C, and cyclin D3, while downregulating p27 protein [104]. In 2014, the same group reached the conclusion that F-2,6-BP mediated the activation of Cdk-1, which regulates p27 ubiquitination, while PFKFB3 silencing inhibited Cdk-1 activity, thereby stabilizing p27 responsible for the G1/S transition (Figure 7). Moreover, they showed that PFKFB3 knockdown induced cell cycle arrest in G1/S in HeLa cells [105]. Similar findings were observed using PFK15 (a small molecule inhibitor of PFKFB3) in gastric cancer cells [77].

In 2018, Shi et al. revealed another possible signaling pathway in which PFKFB3 had an effect on the proliferation of cancer cells. In their study, PFKFB3 knockdown inhibited hepatocellular carcinoma cell proliferation by impairing DNA repair functions, which resulted in G2/M phase cell cycle arrest. It was suggested that this phenomenon might be the outcome of downregulation of ERCC1 expression, which is a protein essential for DNA repair. Downregulation is caused by decreased Akt expression under conditions of PFKFB3 silencing (Figure 7) [106]. The presented duality may explain why some PFKFB3 inhibitors may induce cell cycle arrest in different cell cycle phases. For example, 3PO (first-in-class PFKFB3 inhibitor) is able to induce G2/M phase arrest in Jurkat cells (an immortalized line of human T lymphocyte cells) [12], whereas Kotowski et al. (2020) revealed that the 3-(3-pyridinyl)-1-(4-pyridinyl)-2-propen-1-one (3PO) could also induce G0/1 phase cell cycle arrest in A375 human melanoma cells [107].

PFKFB3 expression also negatively correlates with many proteins involved in epithelial–mesenchymal transition (EMT). Gu et al. (2017) showed that PFKFB3 knockdown not only inhibited the invasiveness of CNE2 human nasopharyngeal carcinoma cells, but also upregulated E-cadherin while downregulating vimentin and N-cadherin levels on the cell surface [66]. Furthermore, it was demonstrated that PFKFB3 siRNA transfection reduced Snail expression and simultaneously upregulated E-cadherin levels in pancreatic cancer cells [78]. These reports underline the important role of PFKFB3 in the proliferation and invasiveness of cancer cells.

Despite fewer studies on the involvement of PFKFB4, this isozyme also seems to contribute to tumor growth. It has been reported that inhibition of its activity reduced cell proliferation and induced cell cycle arrest in the G1/0 phase. In 2017, Li et al. discovered that PFKFB4 mediated the CD44-driven proliferation increase in prostate cancer cells [97]. Furthermore, it was shown in breast cancer cells that PFKFB4 phosphorylated the oncogenic steroid receptor SRC-3, which increased its transcriptional activity and resultant pro-proliferative action [63,108]. Moreover, a negative correlation between the expression of PFKFB4 and histone acetyltransferase GCN5 was demonstrated in thyroid cancer. Knockdown of PFKFB4 inhibited proliferation and invasiveness in IHH-4 thyroid cancer cells, which suggests that the observed effect was mediated by upregulation of GCN5 [109].

Overall, these studies strongly indicate a role of PFKFB3 and PFKFB4 in the invasiveness of cancer cells. However, further (mechanistic) studies are warranted to improve our knowledge regarding this topic, especially in terms of understanding the exact molecular pathways involved.

## 5. Autophagy

Autophagy is a process based on the degradation of cellular molecules and organelles with the aim to produce intracellular energy. It is required for metabolic adaptation in response to various stress stimuli, including oxidative stress, hypoxia, nutrient deprivation, or blockade of glycolysis, to meet energy demands [110,111]. A short insight into autophagy-inducing pathways is depicted in Figure 8.

There are three types of autophagy that lead to cargo degradation: macroautophagy, microautophagy, and chaperone-mediated autophagy. Macroautophagy can be induced under stress conditions to degrade cytoplasmic material and provide metabolites that can be used as an energy source or as substrates for biosynthesis. It relies on de novo formation of cytosolic double-membrane structures (autophagosomes) to transport cargo to lysosomes [112,113,114].

Autophagy induction in tumor cells is related to many factors, including the occurrence of ROS and the unfolded protein response [115]. Its occurrence correlates with diverse genetic polymorphisms and levels of specific proteins such as S100A8/A9, of which the involvement in the induction of autophagy was described by Ghavami et al. in 2010 [116,117].

Recent evidence suggests that autophagy is a potential double-edged sword in cancer, being a tumor suppression mechanism on the one hand and an enabler of tumor cell survival in neoplastic microenvironments on the other. The associations of PFKFB3 and PFKFB4 with autophagy remain unclear. It is very likely that the role of PFKFB3 in inducing autophagy involves ROS through PPP and increased NADPH production. However, some papers present an opposite relationship, where insufficient PFKFB3 activity results in lower ROS availability and reduced autophagy [118].

Initial research in patients with rheumatoid arthritis (RA) showed that lower expression levels of PFKFB3 were associated with a G6P shunt towards PPP, leading to NADPH production and ROS depletion, and, as a result, autophagy inhibition. Forced PFKFB3 overexpression resulted in enhanced autophagic activity (Figure 8) [119].

In HeLa and SK-BR3 cells subjected to nutrient deprivation, ROS production induces phosphorylation of mitogen-activated protein kinase MAPK14 (an essential autophagy mediator), which induces PFKFB3 degradation, shifting metabolism towards PPP, resulting in autophagy inhibition. Therefore, inhibition of MAPK14 results in PFKFB3 upregulation and autophagy activation [120].

Another hypothesis assumes that PFKFB3 inhibition hinders autophagy and cell proliferation by downregulating the 5′ AMP-activated protein kinase (AMPK) signaling pathway [121]. Furthermore, an additional relationship was found, i.e., AMPK became activated during prolonged mitotic arrest. A decrease in AMPK resulted in PFKFB3 phosphorylation, thus increasing PFKFB3 production. Inhibition of AMPK or PFKFB3 led to cell death of breast cancer cells [105].

Additionally, PFKFB3 activity seems to depend on its localization (cellular or nuclear) and is related to redox homeostasis. In both these localizations, these processes were mediated by the AMPK signaling pathway, which appears to play a dual role in the autophagy [122]. Cytoplasmic PFKFB3 was found to strongly promote ATP generation, thereby inhibiting autophagy in renal cell carcinoma (RCC) cells, while nuclear PFKFB3 was associated with autophagy-promoting properties of the same cell line [122].

Resistance to oxaliplatin (used to treat colorectal cancer) often involves the upregulation of autophagy that correlates with increased levels of PFKFB3. The addition of PFK-15 (PFKFB3 inhibitor) to colon cancer cells results in attenuation of autophagy and induction of cytotoxicity. It was hypothesized that the inhibition of PFKFB3 affected biological processes contributing to apoptosis and triggered a glucose shunt towards PPP, further increasing cell susceptibility to apoptosis [70,91,119,123]. It is therefore suggested that combining glycolysis inhibitors with selective inhibitors of autophagy might be a viable therapeutic approach to combat a pro-survival response of cancer cells.

In further studies, dormant breast cancer stem cells (BCSCs) exhibited strong autophagic flux, which resulted in downregulation of PFKFB3. Inactivation of autophagy led to increased PFKFB3 expression, driving the proliferation and outgrowth of BCSCs, thus resulting in reduced self-renewal [124]. As standard chemotherapy inevitably leads to the development of chemoresistance, the observation that PFKFB3 inhibition therapy synergizes with carboplatin and paclitaxel in resistant cell lines of gynecological cancers to reduce tumor weight presents an intriguing therapeutic avenue [82].

The relationship between PFKFB4 and autophagy remains unclear and studies to date show that PFKFB4 induction either up- or downregulates autophagy. Some studies suggest that PFKFB4 depletion decreased the glucose shunt into PPP, which impaired NADPH generation and increased ROS levels, ultimately inducing autophagy [125]. PFKFB4 seems to be positively regulated by endothelial tyrosine kinase, where depletion of either protein decreases autophagy in small cell lung cancer. However, upregulation of PFKFB4 resulted in a poor chemotherapy response [126]. Further studies are warranted to determine the exact molecular pathways involved in the regulation of autophagy by both isozymes.

## 6. Angiogenesis

Angiogenesis is the process of new blood vessel formation from pre-existing vasculature and is characteristic of most solid tumors [127]. It is initiated by local pro-angiogenic factors (e.g., vascular endothelial growth factor; VEGF) which stimulate endothelial cells (ECs) [128]. The blood vessels generated in this process are irregular with gaps between cells, which results in leaking; induction of angiogenesis in the tumor is described in detail in Figure 8. These properties not only accelerate tumor metastasis but also reduce the effective delivery of chemotherapy drugs [129]. While traditional anti-vascular therapies are designed to limit tumor growth by reducing angiogenesis, they might further contribute to hindering the delivery of chemotherapeutics and create a more favorable environment for tumor growth and invasion [130,131,132,133,134]. Moreover, tumors are often able to counteract these targeted therapies through metabolic adaptations; this emphasizes the need for alternative approaches. Angiogenesis requires a considerable amount of ATP as the source of energy (e.g., for the growth of new endothelial tip cells from pre-existing vessels). ATP is mainly provided by glycolysis and mitochondrial respiration; disruption of either of these processes could markedly hamper ATP supply [135]. Tumor endothelial cells (TECs) exhibit a relatively high proliferative activity and rely mainly on glycolysis rather than oxidative phosphorylation as their energy source [136]. Under normal conditions, PFKFB3 expression is already higher in TECs than ECs; further overexpression promotes vessel branching via inhibition of the pro-stalk activity of Notch signaling, which suggests a key role of glycolysis regulation by PFKFB3 in vessel branching (Figure 9) [136].

In cancer, VEGF stimulates PFKFB3 expression and promotes directional migration and filopodia/lamellipodia formation in ECs [137]. PFKFB3 inhibition resulted in suppression of VEGFα protein expression and reduced angiogenic activity [65], whereas PFKFB3 upregulation promoted human umbilical vein endothelial cell (HUVEC) proliferation, migration, and angiogenesis [66].

Targeted inhibition of PFKFB3 by 3PO suppressed vascular hyperbranching and augmented the anti-angiogenic effects of VEGF blockade [138]. As far as mechanistic considerations are concerned, a low-dose administration of 3PO resulted in tightening the vascular barrier by reducing VE-cadherin endocytosis and a reduction in the expression of cancer cell adhesion molecules by downregulating NF-κB signaling. Overall, 3PO reduced cancer cell invasion, intravasation, and metastasis, but failed to affect tumor growth [139].

The effects of different doses of 3PO should be considered when evaluating the therapeutic potential of PFKFB3 inhibitors. While a low dose (25 mg/kg) induced tumor vessel normalization (reducing intravasation and metastasis), a higher dose (70 mg/kg) inhibited cancer cell proliferation and tumor growth. However, it did provoke tumor hypoxia, which resulted in vascular barrier destabilization and promoted tumor dissemination [140].

According to another hypothesis, PFKFB3 may facilitate angiogenesis in oral squamous cell carcinoma by regulating the infiltration of CD163+ tumor-associated macrophages (TAMs), as the expression of PFKFB3 was correlated with CD163 and CD31 [141].

## 7. Targeting PFK-2 Isozymes in Malignancies

### 7.1. Outline of the Development of Inhibitors

Reports on the importance of PFKFB3 and PFKFB4 in cancer development and progression strongly suggest that these isozymes may represent promising targets for new potent personalized therapies in cancer treatment. This prompted research groups to investigate the efficacy of selective inhibitors.

In 1984, Sakakibara et al. identified a binding site of 6-phosphofructo-2-kinase/fructose-2,6-bisphosphatase for Fru-6-P using N-bromoacetylethanolamine phosphate (BrAcNHEtOP) and 3-bromo-1,4-dihydroxy-2-butanone 1,4-bisphosphate [142]. The inhibitory properties of these compounds were later confirmed in both in vitro and in vivo models. However, these inhibitors were not specific and therefore scientists continued to develop novel compounds [32,143]. The first-in-class small molecule PFKFB3 inhibitor, 3-(3-pyridinyl)-1-(4-pyridinyl)-2-propen-1-one, also known as 3PO was synthesized by Clem et al. in 2008 [12]. This compound was computationally identified by screening and docking using ChemNavigator software with a homologous model of the PFKFB3 isozyme, previously generated based on a PFKFB4 crystal structure from rat testes [4,144]. To date, 3PO has been the best-known inhibitor of PFKFB3, chemically belonging to the chalcone group. Its anticancer properties have been demonstrated in experimental models of several types of cancer, including breast cancer [145], ovarian cancer [52], melanoma [107], and bladder cancer [146]. The main factors limiting the potential use of 3PO in clinical trials include poor solubility and difficulty in obtaining sufficiently high concentrations to achieve potency [4]. In 2011, Akter et al. successfully attempted to use a nanocarrier to improve its efficacy in cancer treatment. They conjugated 3PO to micelles prepared from poly(ethylene glycol)-poly(aspartate) [PEG-p(ASP)], which resulted in achieving 2% wt. drug-loading in the nanocarrier polymer. Its favorable properties were observed in Jurkat, HeLa, and LLC cells [147]. In addition, it was shown that cancer cells became more sensitive to microtubule poisons, chemical compounds with the ability to bind to tubulin, thus preventing the formation of microtubules after 3PO treatment [11,105]. Since 3PO is not selective enough, more specific and selective novel PFKFB3 inhibitors were developed in the following years.

In 2011, Seo et al. determined the crystal structure of PFKFB3 and identified new inhibitors such as N4A and YN1. This study not only revealed two inhibitors with increased selectivity for PFKFB3, but was also essential for future targeted drug design due to the extension of knowledge regarding PFKFB3 structure [148].

PFK15, a derivative of 3PO, was another chalcone compound developed to inhibit PFKFB3. The first report of screening, selection, and its impact on cancer cells was published by Clem et al. in 2013 [83]. An increase in binding potency of PFK15 was later achieved by the substitution of the pyridinyl ring with a quinoline ring in 3PO (Figure 9) [4]. This structural modification resulted in an increased selectivity and inhibitory effectiveness (~100-fold), which led to an enhancement of proapoptotic activity compared to 3PO [83]. Due to its modification, PFK15 shows better pharmacokinetic properties, e.g., reduced clearance, higher T1/2, and longer microsomal stability [83]. It was also reported that PFK15 did not inhibit other glycolysis-related enzymes such as phosphoglucose isomerase, PFK-1, PFKFB4, or hexokinase [83].

PFK158, another novel PFKFB3 inhibitor, was proven effective in gynecological cancers [82] and mesothelioma [149]. Moreover, this compound was enrolled in a Phase I clinical trial in patients with advanced solid malignancies [150]. Clinical trials assessing the safety of PFK158 (NCT02044861) were initiated in 2014 and no serious adverse events were reported during the ~one-year follow-up [10]. Once the maximum tolerated dose has been established, Phase II trials of this optimized PFKFB3 inhibitor will also be introduced in leukemia therapy [83].

Since 2014, three novel inhibitors have been developed: compound 26 [151] and PQP [152] and KAN0438757 [153]. The anticancer efficacy was only proven in vitro. KAN0438757 is the most recently (2019) developed PFKFB3 inhibitor and may induce nucleotide incorporation during DNA repair and selectively sensitize transformed cells to the impact of radiation [153].

The recent progress in identifying new drugs targeting PFK-2 isozymes is dominated by compounds inhibiting PFKFB3. This is probably due to a better understanding of this isozyme and its role in cancer cell biology. To date, there has only been one study focusing on PFKFB4 inhibitor design, which reported an anti-proliferative effect of 5-(n-(8-methoxy-4-quinolyl)amino)pentyl nitrate (5MPN) on H460 adenocarcinoma cells. Clearly, these results were promising and justify further investigation of the effects of specific PFKFB4 inhibitors.

The compounds described in this section have a diverse impact on cancer cell metabolism, which is the result of differences in their structures that determine the activity of each compound (IC50 or Ki) and translate into pharmacokinetic properties such as bioavailability and water solubility (Figure 10).

### 7.2. Chemosensitivity, Chemoresistance, and Potential Combined Therapies for Malignancies

Current chemotherapeutic and irradiation protocols target rapidly dividing cells. Targeting glycolysis, a process that is regulated by PFKFB and is crucial for ATP generation in proliferating cancer cells seems to be a promising therapeutic approach with anticancer properties. Combining currently used chemotherapeutics (conventional or tumor pathway-specific agents) with PFKFB3 or PFKFB4 inhibitors is expected to enrich the range of treatment options (Table 2). Despite the exponential development of new drugs, the occurrence of resistance simultaneously increases as well; this emphasizes the importance of implementing drug combinations or adding novel therapeutic agents in an attempt to overcome this hurdle [160]. PFKFB inhibition might prevent disease progression and drug resistance, and even improve progression-free survival (PFS) and/or response rates [10,32]. Another argument in favor of combination therapy with these inhibitors is the association between drug resistance, mitochondrial respiratory defects, and increased glycolysis in cancer cells [11,32,161]. Multiple trials verifying currently approved agents, such as inhibitors of angiogenesis or autophagy, and substances interfering with the electron transport chain or glutamine metabolism, in combination with PFKFB inhibitors are expected to be initiated in cohorts with distinct types of cancers [83]. Liu et al. (2001) suggested that the inhibition of glycolysis markedly sensitizes slow-growing cancers to chemotherapy and irradiation [162].

**Table 2 cancers-13-00909-t002:** Summary of the most relevant studies on the use of PFKFB3 inhibitors in combined therapy.

Inhibitor	Combined Therapy	Type of Neoplasm	Study
3PO	imatinib	chronic myeloid leukemia	Zhu Y, et al. [15]
PFK15	imatinib	chronic myeloid leukemia	Zhu Y, et al. [15]
PFK15	rapamycin	acute myeloid leukemia	Feng Y & Wu L [56]
PFK158	vemurafenib	melanoma	Lu L et al. [10]
PFK158	erlotinib	non-small cell lung cancer cell	Lypova N, et al. [84]
PFK158	antiestrogen	breast cancer	Imbert-Fernandez Y, et al. [58]
PFK15	cisplatin	cervical cancer	Li FL, et al. [163]
PFK158	carboplatin	ovarian cancer	Mondal S, et al. [82]
3PO	paclitaxel	breast cancer	Domenech E, et al. [105]
PFK15/siRNA	oxaliplatin	colon cancer	Yan S, et al. [70]
3PO	VEGF inhibitors	endothelial cells	Schoors S, et al. [138]

#### 7.2.1. Influence on Hematological Malignancies

The JAK2V617F mutation is characteristic of myeloproliferative neoplasms [10,102,164]. STAT5, a transcriptional activator associated with JAK2, induces PFKFB3 expression. The mechanisms underlying this process are still unknown but it has been speculated that downstream effectors of STAT5 play a significant role. Targeted treatment of hematologic diseases with JAK inhibitors leads to the downregulation of PFKFB3 [10]. This reveals the potential benefits of treatment of myeloproliferative neoplasms with PFKFB3 antagonists [10,165]. Combined therapy with glycolysis inhibitors or other currently applied agents has not been introduced to clinical trials yet.

Genetic alterations resulting in BCR-ABL fusion, characteristic of hematologic malignancies, lead to the constitutive activation of tyrosine kinase. Imatinib, an inhibitor of this tyrosine kinase, downregulates glucose uptake in cells with a fusion gene [166]. Despite clinical success, chronic myeloid leukemia resistance to tyrosine kinase inhibitors is still a challenge for clinical hematologists. Interestingly, PFKFB3 was found to be associated with this type of resistance [15]. Therefore, PFKFB3 inhibitors or localized targeted genetic PFKFB3 knockdown might represent a strategy to prevent the development of drug resistance by downregulating glycolysis [10,15,32]. Supporting this notion, treatment with 3PO and PFK15 combined with imatinib and bone marrow transplantation prolonged the survival of mice with chronic myelogenous leukemia (CML) [11,15].

mTOR-mediated upregulation of the PFKFB3 pathway is vital for acute myelogenous leukemia (AML). Accordingly, combined therapy with rapamycin and PFK15 was found highly effective in the inhibition of AML cell proliferation [10,32,56].

#### 7.2.2. Gynecological and Breast Cancers

HER2 expression, which is characteristic of many breast cancers, is associated with increased glucose metabolism mainly via the PFKFB3 pathway [59], suggesting it could represent a potential target of therapy. Indeed, breast cancer cells resistant to anti-HER2 therapy were re-sensitized after inhibition of PFKFB3 and 3PO selectively suppressed the growth of HER2 positive cells through interfering with glycolysis [59]. Additionally, a decrease in glucose uptake after 3PO administration was found in HER2-positive breast tumors [59]. Furthermore, PFKFB3 expression was markedly lower after treatment with the HER2 inhibitor lapatinib [32]. Similar results are expected for combined therapy with HER2 and PFKFB3 inhibitors in trastuzumab-resistant cancer [10].

Cisplatin is a nonspecific chemotherapeutic that interferes with DNA replication through DNA strand crosslinking. However, cancer cells acquire many different mechanisms of drug resistance, such as decreased cellular uptake or increased efflux of drug(s), which require energy. Cisplatin induces PFKFB3 acetylation, which leads to protein overexpression and activation, and markedly enhanced glycolysis [163]; this pathway contributed to cisplatin resistance [163]. Conceivably, glycolysis inhibitors might enhance the efficacy of cisplatin by restoring chemosensitivity [11,139]. PFK15 sensitized cells to platins in a xenograft cancer model, which suggests the possibility of combined therapy [32,163]. PFKFB3 might inhibit resistance to another platin (i.e., oxaliplatin) as well through its influence on autophagy (apart from inhibiting proliferation and migration, oxaliplatin induces autophagy). Following platin administration, cancer cell viability is significantly reduced when autophagy is inhibited, an effect that can be potentiated by the inhibition of PFKFB3 as this restricts autophagy through interfering with its initiation. Other results of the administration of the PFKFB3 inhibitor include reductions in cell viability, proliferation, and migration [70].

Gynecologic malignancies are another potential target for PFKFB3 inhibition. PFK158 caused a reduction in glucose uptake and induction of apoptosis. When combined with carboplatin or paclitaxel, PFK158 acted synergistically but only in chemoresistant human ovarian cancer cell lines (C13, HeyA8MDR). The combination of PFK158 and carboplatin was also efficacious in a mouse model of ovarian cancer, as evidenced by tumor reduction [82]. These preclinical results might soon be followed by clinical trials focusing on chemoresistant gynecological cancers.

#### 7.2.3. Influence on Lung Cancer (NSCLC and SCLC)

Chemoresistance of non-small cell carcinoma (NSCLC) is one of the most important health issues in developed countries as this neoplasm is one of the leading causes of death [167,168]. PFKFB3 expression is a prognostic marker in lung adenocarcinoma [169]. Delivery of shRNA against PFKFB3 enhanced docetaxel activity and significant effects were observed on cell cycle, cell stress, and the balance between survival and apoptosis. The reduction in tumor volume observed in an A549 xenograft model was markedly higher than for docetaxel monotherapy [170]. Antagonism of PFKFB3 and subsequent lower glycolytic activity led to a decrease in cell viability. In addition, tumor cells were less likely to migrate and invade, and cell cycle arrest and apoptosis were observed in these cell lines [169]. The first results are promising, and the next stages of research are expected to verify the usefulness of PFK15 in lung adenocarcinoma therapy.

Epithelial and endothelial tyrosine kinases are crucial for the modulation of small-cell lung cancer (SCLC) chemoresistance through a poorly understood mechanism. A role of PFKFB4 is proposed, as this enzyme is a downstream target of epithelial and endothelial tyrosine kinases. The most important regulatory function of this pathway is the modulation of autophagy and chemoresistance [126]. The potential of combining specific tyrosine kinase inhibitors with PFKFB4 remains to be verified. Nevertheless, PFKFB4 expression can be considered one of the predictive factors for SCLC chemoresistance [126].

#### 7.2.4. Influence on Other Solid Tumors

Overexpression of PFKFB3 followed by glycolysis is responsible for Hepatocellular carcinoma (HCC) resistance to sorafenib. One of the proposed approaches to overcome this resistance is the administration of aspirin. Combined therapy including sorafenib and aspirin induced apoptosis in vitro and in vivo, without causing significant adverse effects [171].

High levels of estradiol, BRAFV600E, or epidermal growth factor (EGF), which are associated with breast cancer, melanoma, and non-small cell carcinoma (NSCLC), respectively, were found to regulate PFKFB3 expression [10,58,172]. Targeted treatment of alterations characteristic of these neoplasms combined with the inhibition of PFKFB3 enhanced the apoptotic pathway and induced cytotoxicity [10]. Combining vemurafenib and PFK158 proved the aptness of this hypothesis in a mouse model, significantly decreasing the tumor size [10]. This combination might improve progression-free survival (PFS) [172]. Several other studies highlighted the therapeutic usefulness of PFK158 combined with estrogen or EGF inhibitors (fulvestrant or erlotinib, respectively) [32,58,84].

CTLA-4 is a glycoprotein expressed on T lymphocytes which regulates initial stages of the immune reaction, thus preventing cell activation. Blocking this process enhances the activity of lymphocytes [173]. Combining anti-CTLA4 agents, such as ipilimumab, and PFKFB3 might simultaneously reinforce the immune reaction to cancer and attenuate cancer cell metabolism. This promising combination is currently being investigated [32,173].

Multidrug-resistant retinoblastoma is another model of chemoresistance in which the participation of PFKFB can be considered. Even though a connection between PFKFB4 expression and resistance to carboplatin, etoposide, or vincristine could not be observed in a study on Y79/EDR cells [174], excluding a role of enhanced glycolysis in acquiring this resistance is premature.

High expression and cellular relocation of PFKFB3 are associated with resistance to radiotherapy, whereas the loss of PFKFB3 enhances radiosensitivity [175]. It is therefore conceivable that the efficacy of radiation treatment can be improved by combining it with PFKFB3 inhibition [32].

Thus far, the emerging results of preclinical studies are promising; the administration of PFK15, 3PO, and other PFKFB3 inhibitors has shown synergy with currently used antineoplastic agents in disrupting glycolysis and eliminating resistance mechanisms. It remains to be evaluated whether the success of preclinical studies will translate into clinical trials.

## 8. Future Perspectives

Reports describing a synergistic or sensitizing effect of PFKFB inhibitors with other chemotherapeutics suggest that these compounds could represent an important adjuvant or additive reagent in future cancer management strategies. However, despite numerous recent discoveries, there is still a strong need to improve our understanding of PFK-2 isozymes, particularly in terms of designing new drugs targeting specific isozymes and associated molecular mechanisms.

## 9. Conclusions

6-Phosphofructo-2-Kinase/Fructose-2,6-Biphosphatase isozymes play a crucial role in cancer biology. Recent discoveries related to PFK-2 isozymes, especially PFKFB3 and PFKFB4, in cancer progression and development, have identified their inhibitors as potent therapeutic agents that could play an important role in future cancer treatment. The large body of relevant research summarized in this review highlights the importance of these isozymes, particularly in cellular processes such as proliferation, migration, apoptosis, and autophagy. The increased efforts made in the research and development of new, more specific drugs interfering with PFKFB activity will provide further insight into the mechanisms that drive cancer pathology in general and ultimately render additional options for effective treatment of different types of cancer.

## Figures and Tables

**Figure 1 cancers-13-00909-f001:**
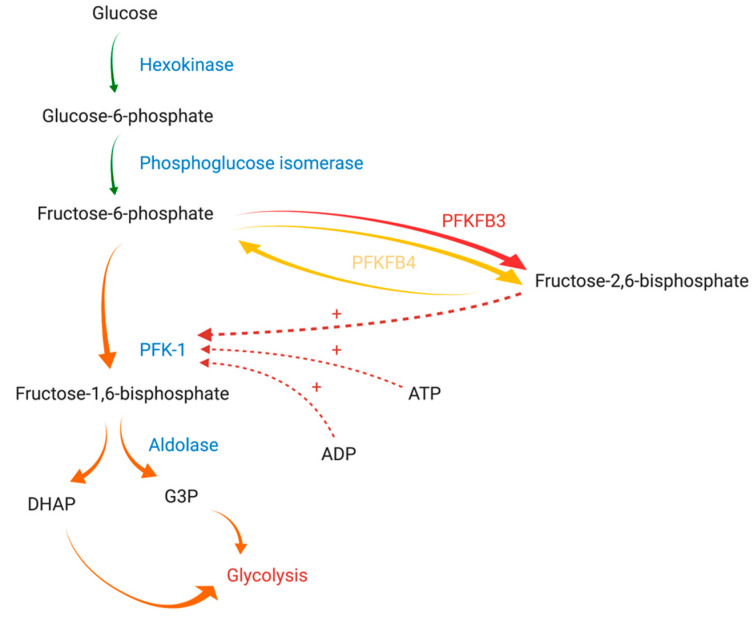
The graphical presentation of PFK-1 regulation by PFKFB3 and PFKFB4 adapted from Yi et al. (2019) and Clem et al. (2008) [11,12]. Diverse arrows colors are used to express the differences between reactions enhancement: (green) normal, (yellow) moderately enhanced, (orange) strongly enhanced, (red) extremely enhanced. Abbreviations: PFK-1 - phosphofructokinase-1; PFKFB3: 6-phosphofructo-2-kinase/fructose-2,6-bisphosphatase isozyme 3; PFKFB4: 6-phosphofructo-2-kinase/fructose-2,6-bisphosphatase isozyme 4; ATP: adenosine triphosphate, ADP: adenosine diphosphate, DHAP: dihydroxyacetone phosphate; G3P: glyceraldehyde 3-phosphate. Created with Https://Biorender.Com/.

**Figure 2 cancers-13-00909-f002:**
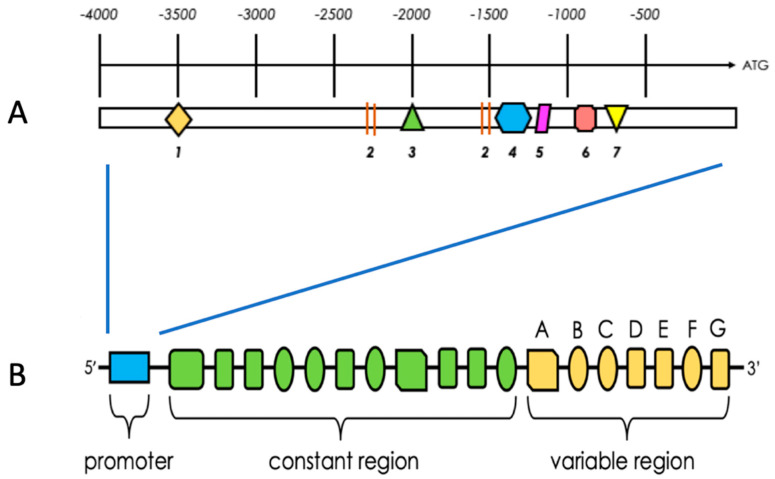
Schematic structure of the 5′ promoter (**A**) of the PFKFB3 gene (**B**). PFKFB3 contains 19 exons subdivided into constant and variable regions. The isoforms of PFKFB3 protein are conditioned by variations in 7 exons (A–G) in the variable region (3’UTR). Numbers in (**A**) represent: 1—progesterone response element, 2—specific protein 1, 3—estrogen response element, 4— early growth response protein (EGR), 5—activating protein 2, 6—hypoxia response element, 7—serum response element. The schematic structures are based on Shi et al. (2017) and Bartrons et al. (2018) [14,32].

**Figure 3 cancers-13-00909-f003:**
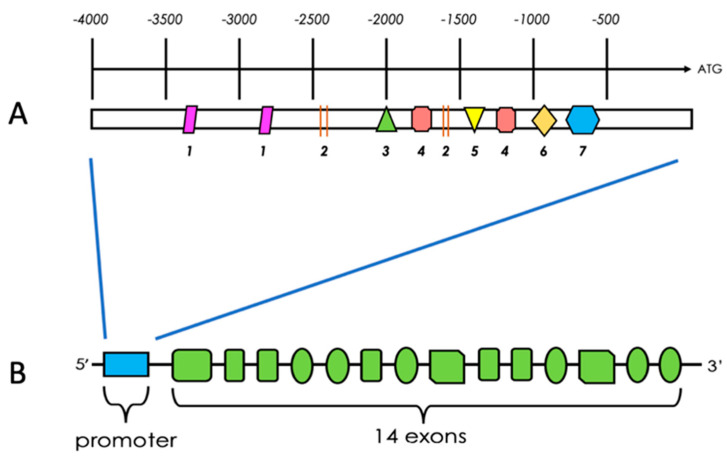
Schematic structure of the 5′ promoter (**A**) of the PFKFB4 gene (**B**). PFKFB4 contains 14 exons. Numbers in (**A**) represent: 1—GRE (glucocorticoid response element), 2—AP-2 (activating protein 2), 3—specific protein 1, 4—TATA box, 5—serum response element, 6—hypoxia response element, 7—ETF. The schematic structures are based on Gomez et al. (2004) [17].

**Figure 4 cancers-13-00909-f004:**
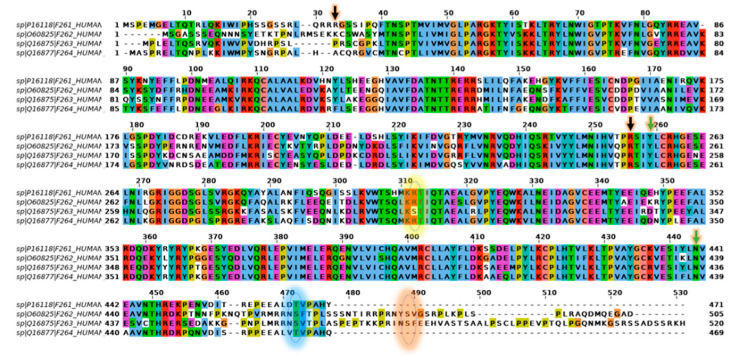
Multi-sequence alignment (MSA) of human PFKFB1, PFKFB2, PFKFB3, and PFKFB4. The black and green arrows with the yellow glow indicate the start and end of N-terminal (2-Kase) and C-terminal (2-Pase) domains, respectively. The yellow glowing circle near amino acid 310 indicates the position where serine 303 of PFKFB3 is located; the blue and red glowing circles indicate where serine 460 (PFKFB3), serine 466, and serine 486 (PFKFB2) are localized. The coloring shows the regions with conserved types of amino acids: blue–hydrophobic amino acids, red–amino acids with a positive charge, magenta–amino acids with a negative charge, green–polar amino acids, pink–cysteine, orange–glycines, yellow–prolines, cyan–aromatic, and white–lack of conservation. The alignment was obtained using ClustalX [41] and is based on sequences deposited in the Uniprot database [42]; the accession numbers are as follows: PFKFB1: P16118-1, PFKFB2: O60825-1, PFKFB3: Q16875-1, PFKFB4: Q16877-1. The figure was prepared using the JalView software [43].

**Figure 5 cancers-13-00909-f005:**
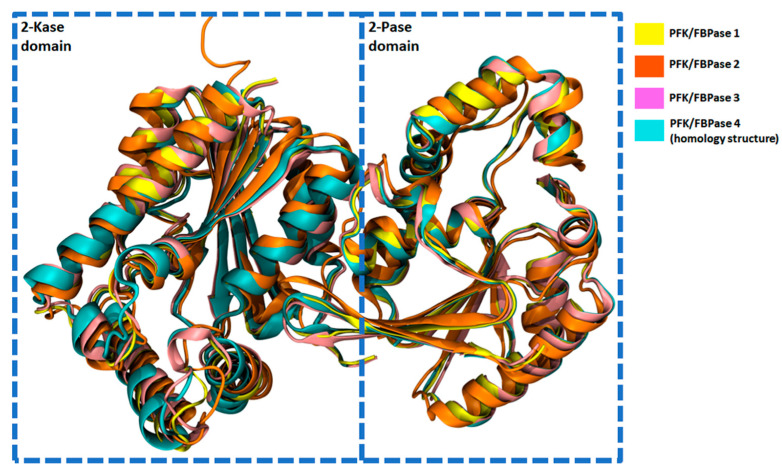
Superimposed structures of human PFKFB1 (yellow), PFKFB2 (orange), PFKFB3 (pink), and PFKFB4 (turquoise); the structures correspond to the sequences shown in Figure 3 and feature the 2-Kase domain (on the left) and 2-Pase domain (on the right). The structures were obtained from the Protein Data Bank (PDB) database [47] with the following codes: 1K6M (PFKFB1), 5HTK (PFKFB2), 6HVI (PFKFB3). The structure for human PFKFB4 was obtained using homology modeling with the Swiss-model [48]; the structure of rat (*Rattus norvegicus*) PFKFB4 (PDB Code 2BIF) shares the vast majority of the human sequence (>96%). The proteins were aligned using PyMol software [46] and the resulting structures were generated using visual molecular dynamics (VMD) software [49].

**Figure 6 cancers-13-00909-f006:**
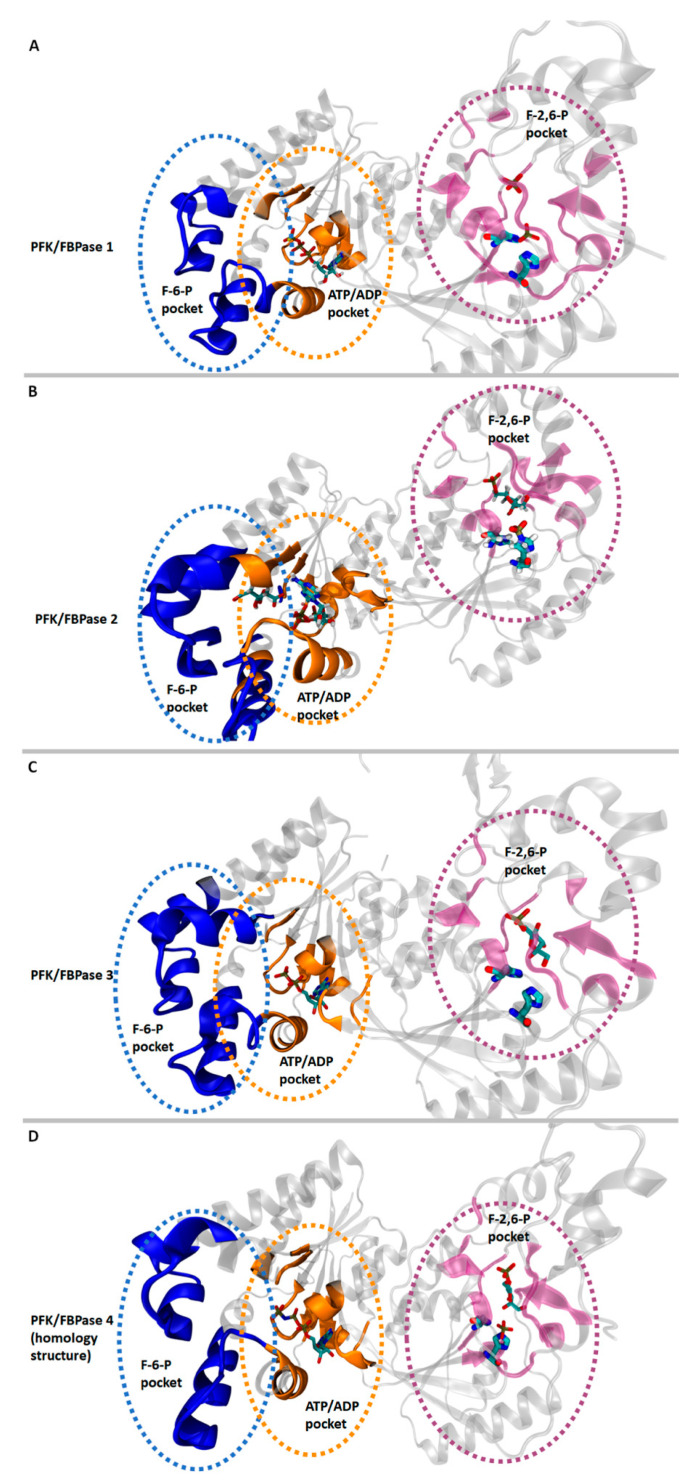
Binding pockets for human 6-phosphofructo-2-kinase/fructose-2,6-bisphosphatase (PFK-2/FBPase-2) isozymes 1 to 4. 2-Kase and 2-Pase domains are shown on the left and right, respectively (as in Figure 5). The subfigures (**A**–**D**) depict PFKFB1, PFBFB2, PFKFB3 and PFKFB4 respectively. Different colors indicate where various substrates/products bind to the crystal structure: the blue-marked pocket is where fructose-6-phosphate (F-6-P) binds (in the case of B, also citrate ion), the orange-marked pocket is where ATP/ADP binds, and the magenta-marked pocket is where F-2,6-P binds. The substrates/products are shown in the pockets if they were present in the crystallographic structure. Furthermore, histidines in the 2-Pase domain, which are responsible for the phosphatase activity, are shown as a thick stick model. The coloring and marking of the pockets are based on the substrates found in the crystal structures and ligand-binding amino acids obtained from the UniProt database [42]; the accession numbers are as follows: PFKFB1: P16118-1, PFKFB2: O60825-1, PFKFB3: Q16875-1, PFKFB4: Q16877-1. The structures were obtained from the PDB database [47] with the following codes: 1K6M (PFKFB1), 5HTK (PFKFB2), and 6HVI (PFKFB3). The structure for human PFKFB4 was obtained using homology modeling with the Swiss-model [48]; the structure of rat (*Rattus norvegicus*) PFKFB4 (PDB Code 2BIF) shares the vast majority of the human sequence (>96%). The positions of the substrates were also derived from the 2BIF crystallographic structure. The protein structures were generated using VMD software [49].

**Figure 7 cancers-13-00909-f007:**
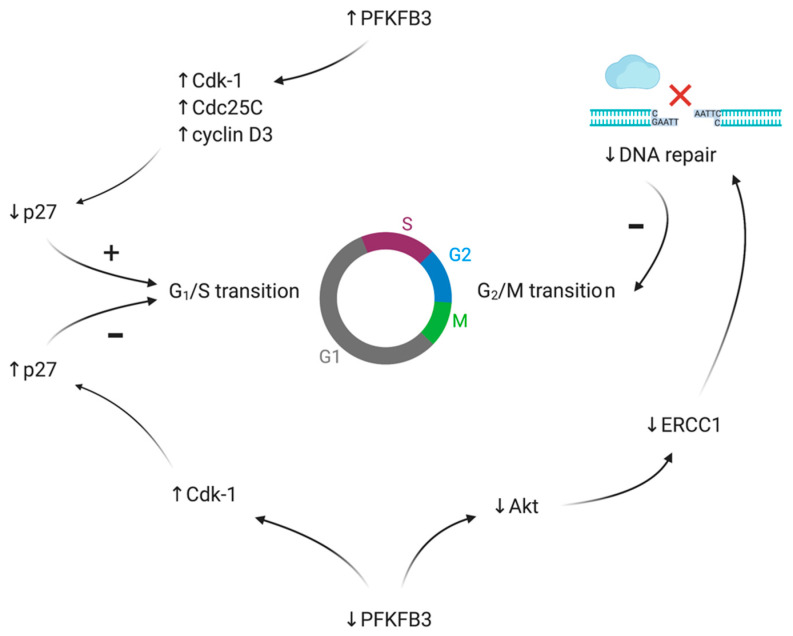
Graphical presentation of the PFKFB3 impact on cycle progression. Abbreviations: PFKFB3: 6-phosphofructo-2-kinase/fructose-2,6-bisphosphatase isozyme 3, Cdk-1: Cyclin-dependent kinase 1, Cdc25C: M-phase inducer phosphatase 3, cyclin D3: G1/S-specific cyclin-D3, p27: Cyclin-dependent kinase inhibitor 1B, Akt: Protein kinase B (PKB), ERCC1: Excision Repair Cross-Complementation Group 1. Created with Https://Biorender.Com/.

**Figure 8 cancers-13-00909-f008:**
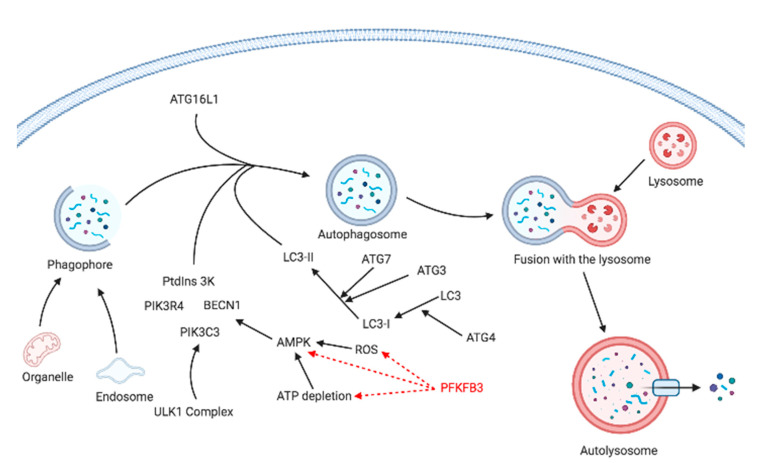
A brief overview of the autophagy pathway, which is a catabolic process that includes a specific intracellular cargo, such as organelles or endosomal contents that are intended for degradation. The autophagy process is initiated by an extracellular stimulus or cargo recognition that prompts the formation of the phagophore. The cargo is then engulfed within a double-membrane vesicle (an autophagosome). Increased initiation to engage phagophores for autophagy involves the activation of the ULK1 complex that further induces the nucleation complex, which includes PtdIns3K, PIK3, and BECN1. Later, LC3 is conjugated to the phagophores and is responsible for their maturation and elongation. Upon maturation, the autophagosome fuses with a lysosome, with its content being released and degraded by lysosomal enzymes. Recent evidence suggests that PFKFB3 induces autophagy through increased Nicotinamide adenine dinucleotide phosphate (NADPH) production. However, some dissertations suggest an opposite relationship—insufficient PFKFB3 activity results in lower Reactive Oxygen Species (ROS) availability and reduced autophagy. Mitochondrial damage increases ROS and decreases cellular ATP levels. ROS may activate AMP-activated protein kinase (AMPK), which positively regulates autophagy through phosphorylation of BECN1. Abbreviations: ULK1 complex—unc-51 like autophagy activating kinase 1 complex, PIK3C3—phosphatidylinositol 3-kinase catalytic subunit type 3, PIK3R4—phosphatidylinositol 3-kinase regulatory subunit 4, BECN1—Beclin 1, PtdIns 3K—phosphatidylinositol 3-kinase, LC3—Microtubule-associated proteins 1A/1B light chain 3B, LC3-I—cytosolic form of LC3, LC3-II—lipid modified form of LC3, ATG—autophagy-related protein. Created with Https://Biorender.Com/.

**Figure 9 cancers-13-00909-f009:**
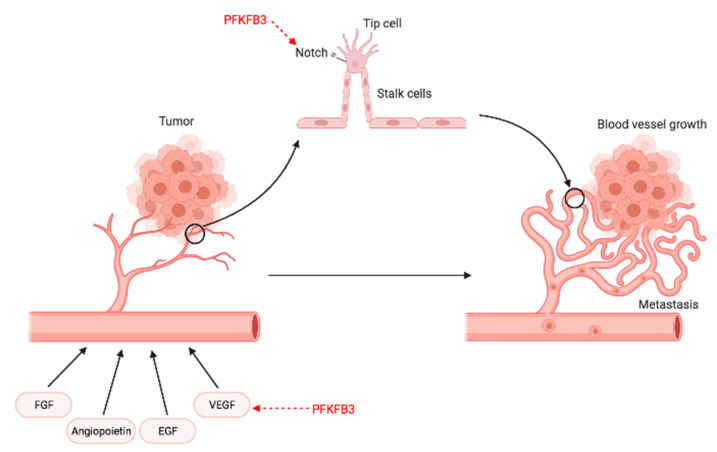
Schematic representation of angiogenesis, a process in which new blood vessels are developed from pre-existing vessels, allowing for tumor progression and metastasis. During angiogenesis, vascular permeability is increased in the existing vessels, which allows for extravasation, degradation of the extracellular matrix, and release of sequestered growth factors. When exposed to stimulating factors (such as VEGF, FGF, and EGF), endothelial cells proliferate, migrate, and form primary sprouts. Vessel sprouting by migrating tip and proliferating stalk cells is controlled by genetic signals, such as Notch. Further proliferation, followed by the synthesis of a new basement membrane and maturation, leads to assembly of lumen-bearing cords. PFKFB3 levels affect angiogenesis through mediating VEGF activity, where PFKFB3 upregulation results in enhanced VEGF activity. Moreover, silencing of PFKFB3 impairs angiogenesis, while PFKFB3 overexpression overrules the pro-stalk activity of Notch. Abbreviations: FGF—fibroblast growth factor, EGF—endothelial growth factor, VEGF—vascular endothelial growth factor. Created with Https://Biorender.Com/.

**Figure 10 cancers-13-00909-f010:**
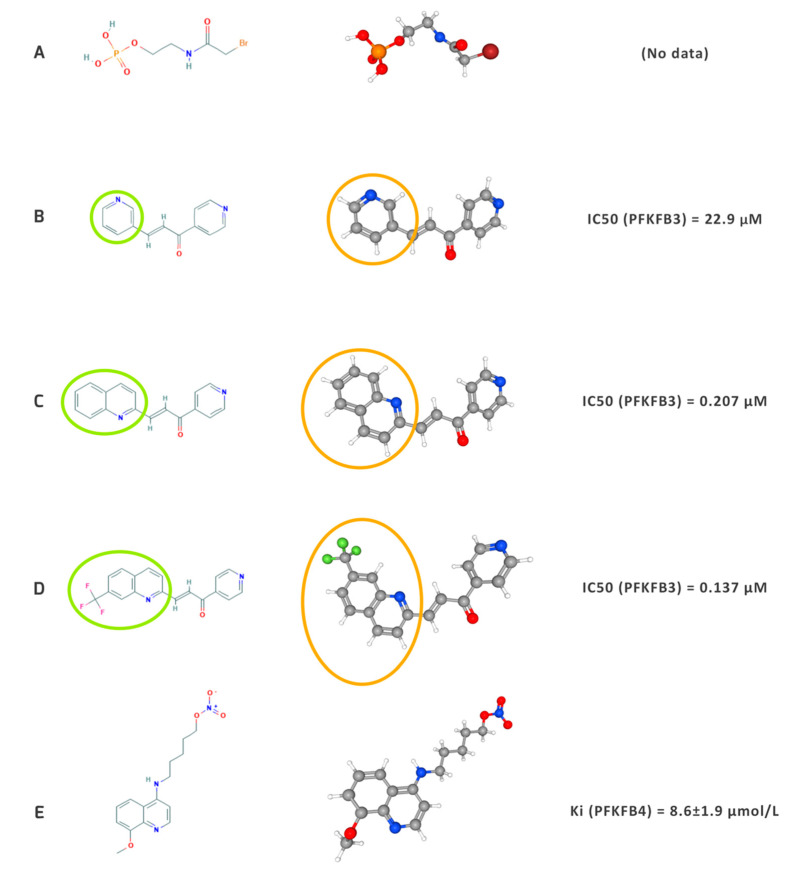
Structural models (on the left), 3D models (in the middle) and IC50 or Ki values of the most important PFKFB3 (**A**–**D**) and PFKFB4 (**E**) inhibitors to date. The models were obtained from PubChem: (**A**)—N-Bromoacetylethanolamine phosphate [154], (**B**)—3-(3-pyridinyl)-1-(4-pyridinyl)-2-propen-1-one (3PO) [155], (**C**)—PFK15 [156], (**D**)—PFK158 [157], (**E**)—5MPN [158]. The circles illustrate the differences in the chemical (green circles) and 3D structure (orange circles) between the 3PO (**B**), PFK15 (**C**), and PFKF158 (**D**)–consecutive derivatives chalcone compounds. IC50 and Ki values were obtained from Wang et al. (2020) [4] and Chesney et al. (2015) [159].

## Data Availability

No new data were created or analyzed in this study. Data sharing is not applicable to this article.

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
