# Peer review of "Role of PFKFB3 and PFKFB4 in Cancer: Genetic Basis, Impact on Disease Development/Progression, and Potential as Therapeutic Targets"

_cancers, 2021, doi:10.3390/cancers13040909_

Round 1

Reviewer 1 Report

This is a comprehensive review which will be a good resource for researchers in the area of metabolism and cancer. It is well written and very informative.

Just a few questions and suggestions:

Why only draw the gene structure for PKF2? It would be best to draw all or at least explain why this particular one?

In Line 264, regarding PKF isoform regulation in hypoxia, the authors should rephrase stating that these isoform is a HIF target and hence levels are increased in hypoxia. As it stand, it looks like HIF activates the enzyme, which is not correct.

Author Response

This is a comprehensive review, which will be a good resource for researchers in the area of metabolism and cancer. It is well written and very informative.

            First, we strongly want to express our gratitude for your review, consideration, favourable assessment and very valuable suggestions.

  • Why only draw the gene structure for PKF2? It would be best to draw all or at least explain why this particular one?

Answer: We appreciate your suggestions regarding the illustration of all PFKFB1-4 genes in Figure 2 (line 158). Implementing the figure describing all the four genes, as you suggested, would probably be compatible with other figures, unfortunately, the data describing PFKFB1 and PFKFB2 were in our opinion not enough to safely draw their structure, however, the research on PFKFB4 and its promoter in a human was adequate and data was enough. Therefore, we decided to draw the PFKFB4 gene and its promoter and this figure was added to our manuscript as the additional Figure 3 (line 179). I hope that after addressing this issue the manuscript becomes more consistent and informative.

  • In Line 264, regarding PKF isoform regulation in hypoxia, the authors should rephrase stating that these isoform is a HIF target and hence levels are increased in hypoxia. As it stand, it looks like HIF activates the enzyme, which is not correct.

Answer: We completely agree that the sentence describing the impact of HIF-1alfa on the PFK-2 isozymes, might be confusing and not clearly stated. As you suggested we paraphrase some statements and highlight that this regulation is made through the influence of the expression of PFKFB isozymes. We hope that after this change this mechanism is described with more accuracy (line 154 and 281).

Finally, the English language was checked previously by the experienced English editor from Poland. We understand that some of the sentences might not meet the expected level of language for the journal. Therefore, to address this issue and improve its quality the whole manuscript has been checked carefully again by another qualified English editor specialized in the science English writing from Canada. We hope that this double proofreading improved the quality, correctness as well reading flow.

            Summarizing, once again we are very grateful for your all valuable remarks and hope that after implementation of the aforementioned changes our manuscript will meet the requirements to be accepted for publication in ‘Cancers’ Journal.

Reviewer 2 Report

Overall, this is an exhaustive and informative review of the subject.

I have a few comments that I believe may improve the manuscript.

  1. It is apparent that an English speaker did not write the entire manuscript. There are many awkward sentences and grammatical errors. Sometimes the wrong words are chosen. Some examples: line 255 – “Therefore, it is possible to be another reason for such a high 2- 256 Kase/2-Pase activity ratio of PFKFB3”, line 200 the choice of “outburst” when the authors mean elucidation, characterization, or description (all better choices).

In particular, the initial Simple Summary is not well written and retracts from an otherwise excellent review.

  1. In the introduction line 68, the authors may consider adding a reference (and a line or two of text to discuss). This reference is Cell: 2020 Rigel et al and describes the preferential glycolysis pathway in cancer cells and cancer associated immune cells within the tumour.
  2. A few comments on figures:
    1. Figure 1 – Is there a difference the authors want to emphasize with the use of different coloured arrows? This is not clear and if there is no difference, they should not be multicoloured just for esthetics.
    2. Figure 6 – Not very informative and much too general in its scope. The figure should mirror the information in the text. As it is, the figure does not describe the text.
    3. Figure 7 and Figure 8 – in both cases, the role of PFKFB is not apparent. These figures are fine to describe a cellular process but there is no connection to the topic of the paper. In the text, there is relevant information about how these processes are affected by PFKFB. This should appear in the figures.
    4. Figure 9 is unnecessary and does not contribute to a better understanding of the text.

Author Response

Overall, this is an exhaustive and informative review of the subject.

I have a few comments that I believe may improve the manuscript.

We are very grateful for your review and your advice. We appreciate your valuable remarks on our manuscript, thus we could improve upon it.

  • It is apparent that an English speaker did not write the entire manuscript. There are many awkward sentences and grammatical errors. Sometimes the wrong words are chosen. Some examples: line 255 – “Therefore, it is possible to be another reason for such a high 2- 256 Kase/2-Pase activity ratio of PFKFB3”, line 200 the choice of “outburst” when the authors mean elucidation, characterization, or description (all better choices).

In particular, the initial Simple Summary is not well written and retracts from an otherwise excellent review.

Answer: English corrections were checked previously by the experienced English editor from Poland. We understand that some of the sentences might not meet the expected level of language for the journal. Therefore, to address this issue and improve its quality the whole manuscript has been checked carefully again by another qualified English editor specialized in the science English writing from Canada. We hope that this double proofreading improved the quality, correctness as well reading flow to meet the requirements of the Cancers Journal readers.

  • In the introduction line 68, the authors may consider adding a reference (and a line or two of text to discuss). This reference is Cell: 2020 Rigel et al and describes the preferential glycolysis pathway in cancer cells and cancer associated immune cells within the tumour.

Answer: In terms of adding the reference in line 68, we also want to express our gratitude for suggesting this article for discussion in our manuscript. Unfortunately, we could not find an article that fits the description mentioned by you (Rigel et al., Cell, 2020). Nevertheless, we carefully looked for all articles of ‘Rigel et al’ authors and found one (Rigel J. Kishton et al., Cancer Journal, 2015) which fit your description of the manuscript and simultaneously enrich the introduction in our paper.

  • A few comments on figures:

Answer: Below we described changes in the Figures.

Figure 1. The different colours of the arrows were used deliberately. In terms of the distinct colour of arrows for PFKFB3 and PFKFB4, we wanted to express the difference in the kinase/phosphatase ratios - about 730-fold higher for PFKFB3 than for PFKFB4. When it comes to the arrows between the individual product compounds in the glycolysis, we wanted to highlight the increased activity of the further reactions after PKF-I activation in the illustrated mechanism. The proper note with its explanation was also implemented in the figure description to avoid ambiguities (line 74).

Figure 6 (Now Figure 7). We appreciate your suggestions and feedback regarding this figure. Therefore, we decided to extend the illustration of the description of additional data to make this figure more informative and suitable regarding the major flow of the manuscript (line 342).

Figure 7 (Now Figure 8). We have expanded the figure in order to present the potential role of PFKFB3 in the regulation of the autophagy process. The figure now includes the AMPK pathway, which is activated by ROS and ATP depletion. For a better visibility, we highlighted PFKFB3 and its targets in red (line 394). The adequate description below the figure was also added (line 402).

Figure 8 (Now Figure 9). We have adjusted the figure to focus more on the potential role of PFKFB3 in angiogenesis - it now involves the mechanistic effect of PFKFB3 on angiogenesis (line 479). The adequate description below the figure was also added (line 483).

Figure 9 (Now Figure 10). In order to improve the informative validity of this figure we marked and compared the chemical structure changes between illustrated compounds where it was possible (line 593). The adequate description below the figure was also added (line 596).

Summarizing, once again we are very grateful for your all valuable remarks and hope that after implementation of the aforementioned changes our manuscript will meet the requirements to be accepted for publication in Cancers Journal.

Reviewer 3 Report

In this review paper, the authors comprehensively described the role of PFKFB3 and PFKFB4 in cancer. They described the sequence and structure similarities and differences of four different PFK2 genes. Also, they described the gene expression of PFKFB3 and PFKFB4 in numerous cancer types and explained their roles in proliferation, invasiveness, migration, autophagy, and angiogenesis. Finally, they described various ways to target PFK-2 isozymes in several malignancies. The authors have compiled and presented several aspects of PFK2 genes, including glycolysis, one of the important hallmarks of cancer. This review is very useful and will be of interest to many readers of Cancers.

Below are some suggestions and minor corrections that will hopefully improve this interesting review.

  1. From Ensembl, PFKFB1 has four splice variants, namely PFKFB1-201, PFKFB1-202, PFKFB1-203, and PFKFB1-204. Please update the PFKFB1 section (line 130)
  2. Line 150, change Glut1 to SLC2A1 (official gene symbol for GLUT1).
  3. Section 3 on PFK-2/FBPase-2 isozymes and diseases - since the main focus of this review is on cancers, section 3 on other diseases looks out of place. It's better to remove this section to maintain the flow of the review.
  4. Since these isozymes are involved in glycolysis, it will be useful to describe their association with cancer metabolism, another important hallmark of cancer.
  5. Are there any genetic mutations or copy number alterations present in these genes?

Author Response

In this review paper, the authors comprehensively described the role of PFKFB3 and PFKFB4 in cancer. They described the sequence and structure similarities and differences of four different PFK2 genes. Also, they described the gene expression of PFKFB3 and PFKFB4 in numerous cancer types and explained their roles in proliferation, invasiveness, migration, autophagy, and angiogenesis. Finally, they described various ways to target PFK-2 isozymes in several malignancies. The authors have compiled and presented several aspects of PFK2 genes, including glycolysis, one of the important hallmarks of cancer. This review is very useful and will be of interest to many readers of Cancers.

Below are some suggestions and minor corrections that will hopefully improve this interesting review.

  • From Ensembl, PFKFB1 has four splice variants, namely PFKFB1-201, PFKFB1-202, PFKFB1-203, and PFKFB1-204. Please update the PFKFB1 section (line 130)

Answer: First of all, we strongly want to express our gratitude for your review and valuable remarks.

            In terms of the splice variants of the PFKFB1. As you suggested we add the note and describe the splice variants from the Ensembl database (line 137).

  • Line 150, change Glut1 to SLC2A1 (official gene symbol for GLUT1).

Answer: The suggestion considering changing the name of the gene - ‘Glut 1’ for the correct one ‘SLC2A1’ was extremely helpful. We corrected it (line 156).

  • Section 3 on PFK-2/FBPase-2 isozymes and diseases - since the main focus of this review is on cancers, section 3 on other diseases looks out of place. It's better to remove this section to maintain the flow of the review.

Answer: In your review, you mentioned that Section 3 describing the PFKFB3 and PFKFB4 in diseases was redundant. I agree with your remark and this section was removed.

  • Since these isozymes are involved in glycolysis, it will be useful to describe their association with cancer metabolism, another important hallmark of cancer.

Answer: We provided additional sentences in paragraphs 1 (line 108) and 4 (line 331) to highlight that glycolysis is a hallmark of cancer. Moreover, the association between glycolysis and cancer is mentioned in paragraphs 2.7.3., 3.2., and 4.

  • Are there any genetic mutations or copy number alterations present in these genes?

Answer: Thank you also for the question about the mutations in the PFKFB1-4 gene. Up to date, this data is missing and this field is poorly unknown. As it was mentioned in the manuscript text - ‘Finally, mutations in PFK-2 isozymes have been detected in several cancer tissue samples, especially in endometrial cancer, colorectal cancer and melanoma (our preliminary data, not published yet)’ - this is the interest of our current research. We performed and obtained the preliminary data. However, further experiments are performing and we hope to be able to share and publish them in the following years.

Finally, the English language was checked previously by the experienced English editor from Poland. We understand that some of the sentences might not meet the expected level of language for the journal. Therefore, to address this issue and improve its quality the whole manuscript has been checked carefully again by another qualified English editor specialized in the science English writing from Canada. We hope that this double proofreading improved the quality, correctness as well reading flow.

Summarizing, once again we are very grateful for your all valuable remarks and hope that after implementation of the aforementioned changes our manuscript will meet the requirements to be accepted for publication in ‘Cancers’ Journal.